# Conceptual framework of applying CoI-based blended learning approach to enhance students' vocabulary and vocabulary learning motivation

Qiu Chuane[ID]*

School of Foreign Languages, YiBin University, YiBin, Sichuan Province, China

* 2856126876@qq.com

## Abstract

In China, secondary school students enrolled in vocational stream frequently have difficulties in mastering the English language. The mechanical traditional way of teaching and their lack of vocabulary toward the target language often demotivate them in learning. This quasi-experimental study aims to investigate the use of CoI-based blended learning approach to enhance students' Vocabulary Test Scores (VTS) and Vocabulary Learning Motivation (VLM) in Chinese secondary vocational English as a Foreign Language (EFL) classrooms. Explanatory sequential mixed methods approach was applied by this study. Using purposive sampling method, a total of 86 first-year students at a Technician College in Chongqing, China was allocated to an experimental group (n = 44) and a control group (n = 42). Instruments involving a researcher-developed vocabulary test and VLM questionnaire were used to gather quantitative data, whilst a semi-structured interview was used to gather qualitative data. Both the quantitative and qualitative findings revealed that CoI-based blended learning approach had the potential of enhancing students' VTS and VLM. Consequently, this research may provide other EFL practitioners with a CoI-based Blended Learning Conceptual Framework for enhancing students' VTS, VLM and insights into the potential of blended teaching supplements through the lens of the CoI framework.

## Introduction

In 2018, the Chinese Ministry of Education (MOE) issued the "National Vocational Education Reform Plan", explicitly highlighting the importance of vocational education on par with general education. However, the reality reveals a significant and growing imbalance in the enrolment ratio between these two streams of education. To address and prevent this worsening imbalance, the Chinese MOE issued the "Notice on the Enrolment in Secondary Vocational Schools" in March 2021, mandating equal enrolment ratios between secondary vocational schools and general high schools.

**Data availability statement:** All relevant data have been uploaded as supplementary files.

**Funding:** The author(s) received no specific funding for this work.

**Competing interests:** The authors have declared that no competing interests exist.

Additionally, in accordance with the "Fourteenth Five-Year Plan and the Outline of the 2035 Vision Goals", China plans to establish the Classified College Entrance Examination (CCEE) system for vocational education and develop numerous high-level undergraduate vocational majors, colleges, and universities. In this type of setting, the importance of English as a compulsory subject for the CCEE in vocational education will steadily increase over the coming decades.

In EFL education, vocabulary is generally regarded as the cornerstone and key component [1]. However, in the traditional method of cramming teaching and rote learning, vocabulary acquisition is often undervalued and attributed to limited time [2]. Nation and Meara [3] noted that the heavy burden of English vocabulary learning often demotivates vocational students, as mastering multifaceted vocabulary knowledge requires frequent and continuous input. Students in China's vocational pathway of secondary education also struggle to keep up with their English language studies. This is due to a weak vocabulary foundation and a lack of motivation in learning it. In this sense, students are often perplexed by exam questions and get very low test scores, hindering their progress in learning English [4].

In recent years, the adoption of blended learning has surged, owing to its flexibility and the integration of digital tools that foster both independent and collaborative learning [1,5,6]. According to [7], blended learning is a technology-based teaching model that combines classroom instruction with technological applications. It seeks to combine different learning methods efficiently in order to motivate students to participate both inside and outside of the classrooms [1]. Pointed out by [1], blended learning is a flexible and meaningful learning method, allowing students to learn at any time and location without any constraints. As a result, students are able to study at their own pace and receive immediate feedback regarding their performance.

According to [8], social-cultural constructivism views education as a collaborative reconstruction of learning experiences. On this basis, research on blended learning also highlights the importance of community and interaction to promote students learning outcomes [9]. Collaboration and teamwork are emphasized [10] among both teachers and students by using communication technologies to deliver knowledge and interact with each other [11]. In this regard, blended learning includes more than just combining online and traditional instruction, it also connects individual students through collaborative inquiry communities [10]. This is in accordance with the idea of the Community of Inquiry (CoI) framework developed by [12], which is popular in online and blended learning environments [13].

Over the years, CoI framework has been applied to diverse disciplines and educational environments, including influence on students engagement and learning outcomes [14], perceived learning [9], online discussions [15], oral skills [16], and virtual reality (VR)-based integrated learning [17]. However, there is a lack of clear explanations as to how blended learning interventions using CoI framework may affect students EFL Vocabulary Test Scores (VTS) and Vocabulary Learning Motivation (VLM) both globally or in Chinese secondary vocational EFL classrooms. Therefore, this study seeks to answer the following research questions:

1. What is the significant difference of using CoI-based blended learning and face-to-face approaches intervention in:

1A1: students' VTS in terms of their pre-test and post-test scores ?

1A2: students' VLM in terms of their pre-intervention and post-intervention scores ?

2. What are student's perceptions of using CoI-based blended learning approach for enhancing their VTS and VLM in the context of Chinese secondary vocational schools' EFL education?

## Literature review

### Community of Inquiry framework

Community of Inquiry (CoI) is generally defined as a collaborative environment or process where students can actively involve and achieve personally meaningful learning [18,19]. It is a major theoretical framework for online and blended learning environments [13,20]. Using this framework, students can learn within a community by interacting dynamically with its three core elements: Teaching Presence (TP), Social Presence (SP), and Cognitive Presence (CP) [21].

Garrison et al. [19] classified Teaching Presence (TP) into three sub-components: design and organization, facilitating discourse, and direct instruction. It can be considered a function of course facilitation and design, which includes identifying, organizing, and presenting course content, course design, curriculum outlines, teaching methods, and teaching behaviors [22]. Mielikäinen and Viippola [23] highlighted that TP encompassed not merely the teacher's physical presence but also the students' perceptions of the teacher's instructional behaviours and effectiveness. Through shared accountability and oversight in the educational process, students can assume the role of educators, facilitating and monitoring their own learning through collaboration [10]. It assists students in establishing learning objectives, formulating learning frameworks, coordinating learning activities, and imparting knowledge [24]. TP enhances learning performance by facilitating cognitive thinking and social contact among students [24], while also promoting students' engagement, progress management, and the systematic organisation of learning [10]. It has been recognised as a crucial factor that affects students learning results and motivation [25].

According to [26], Social Presence (SP) comprises three sub-components: affective expression, open communication, and group cohesion. SP is defined as the capacity for students to engage and participate within the community, communicate intentionally, and cultivate interpersonal relationships with educators and peers [10]. In this environment, establishing a harmonious learning environment is deemed essential [27], enabling students to authentically express themselves through emotional articulation, open communication, and collaborative cooperation [28]. Online contacts with peers may augment mutual trust among students who experience loneliness or isolation in the new learning environment [29]. Prior studies have demonstrated that increased SP correlates with enhanced participant engagement and a higher likelihood of students' attaining motivation, happiness, and learning outcomes [30,31].

Cognitive Presence (CP) is recognized as the core of discourse to be examined [32]. It transpires through a triggering event, exploration, integration, and resolution [33]. CP refers to students ability to develop cognitive skills to comprehend knowledge conveyed through interactions or communications [24,27]. Innovative or challenging learning content may increase students' curiosity and promote their participation in interactive learning activities for further inferences, analyses, and computations [27]. Consequently, educators must establish explicit participation criteria, adaptable technology integration, and organized discussion forums to enhance students' cognitive engagement [34]. Research indicates that elevated cognitive talents may augment students' initiative, therefore fostering robust motivation and a strong belief in learning [35]. Higher-order skills such as metacognition, critical thinking, and creativeness might also be increased and improved by cognitive abilities [36]. Besides, CP is proven to be closely associated with students' learning performance in different professional backgrounds [31].

## Community of Inquiry and learning motivation

Learning motivation influences students' learning behaviors and guides them towards particular academic objectives [37]. Self-determination theory (SDT) posits that an individual's desire to learn during an activity is contingent upon the extent to which the learning environment fulfills the three fundamental psychological requirements of autonomy, competence, and relatedness [38]. According to [39], these three fundamental psychological needs are significantly correlated with the three dimensions of the Community of Inquiry (CoI) framework: Teaching Presence (TP), Social Presence (SP), and Cognitive Presence (CP).

Comprehensive studies have established a substantial correlation between Teaching Presence (TP) and students' motivation [40], especially within the context of Self-Determination Theory (SDT). Autonomy support is augmented when educators employ structured instructional frameworks, offer avenues for students' choice, and promote self-directed learning [41]. By providing timely, substantive feedback and exhibiting passion, educators create an environment that empowers students to take initiative, thus nurturing intrinsic motivation [42]. Moreover, active listening, empathy, and personalized support enhance students sense of agency, consistent with autonomy as a fundamental catalyst for engagement [43]. Secondly, TP enhances competence via constructive feedback, explicit expectations, and scaffolded learning experiences [44]. When educators create engaged and intellectually stimulating sessions, students cultivate mastery beliefs and self-efficacy [45]. A well-organized educational setting, along with formative evaluations, allows students to monitor their advancement, hence enhancing their perceived competence [46]. Ultimately, TP enhances relatedness by cultivating a secure, inclusive, and cooperative classroom environment. Educators who demonstrate positive regard, accessibility, and emotional support foster a sense of value and connection among students [46]. The presence of social interaction in education, facilitated by responsive communication and community-building initiatives, improves relationships among peers and between instructors and students, thereby satisfying students' demand for belonging [43]. This sense of connection is essential for enduring motivation, since it reduces disengagement and enhances emotional well-being [47].

Empirical research indicates that Social Presence (SP) is essential for satisfying learners' psychological needs for autonomy, competence, and relatedness, thus augmenting intrinsic motivation and academic engagement [48]. First, SP enhances autonomy by promoting student-centered learning environments in which learners engage actively in decision-making and collaborative activities [49]. Students participating in peer discussions, group projects, or self-directed activities gain increased control over their learning, corroborating SDT's claim that autonomy-supportive environments augment motivation [47]. Moreover, interactive and dialogic pedagogical approaches, including peer evaluations and collaborative problem-solving, enhance learners' feeling of agency, diminishing dependence on external regulation and fostering self-determined behavior [48]. Second, the concept of competence is enhanced by SP, facilitating meaningful interactions that affirm learners' capacities and support skill development [50]. Collaborative exercises, including group discussions and peer feedback sessions, enable students to assess their comprehension, enhance their knowledge, and obtain constructive feedback from peers and instructors. These interactions enhance subject-matter expertise and foster metacognitive and self-regulatory skills, vital for enduring academic achievement [51]. When educators facilitate supportive and organized social interactions, students cultivate enhanced self-efficacy, viewing themselves as competent participants in the learning community [52]. The most immediate impact of SP is on relatedness, since it fulfills the fundamental human need for connection and belonging [53]. A strong sense of community in educational settings enhances students emotional engagement and reduces feelings of isolation [54]. Small-group discussions and cooperative learning tasks, foster students' trust and mutual support [54]. This interactive and socially enriched environments augment intrinsic motivation by validating students identities as legitimate participants in academic discourse [55].

Studies have repeatedly shown a significant beneficial correlation between Cognitive Presence (CP) and students' motivation [56,57]. CP promotes autonomy by facilitating self-directed inquiry and intellectual exploration [58]. Students participate in higher-order cognitive processes, including critical reflection and knowledge application, achieve enhanced congruence between their learning activities and personal values [59]. CP's problem-solving aspect enables learners to

exercise autonomous judgment, make significant decisions regarding their learning methodologies, and cultivate a sense of ownership over their educational experience [60]. The correlation between CP and competency is well-documented in the literature. By engaging consistently in critical discourse and using knowledge practically, students build subject-matter competence and metacognitive skills [61]. The iterative process of inquiry, reflection, and problem-solving intrinsic to CP offers ongoing chances for students to showcase and enhance their abilities [62]. When educational settings offer explicit learning objectives and formative assessment opportunities, students report increased emotions of competence and self-efficacy [63]. The predominantly cognitive aspect of persistent critical thinking and knowledge production also promotes relatedness via collaborative meaning-making [58]. The social aspect of CP arises during collaborative problem-solving tasks in which students participate in collective reflection and analysis [64]. This shared cognitive endeavor fosters a sense of intellectual community, fulfilling students need for belonging in academic environments [61]. The reflective elements of CP allow students to relate their learning experiences to wider social contexts, thus augmenting their sense of purpose and affiliation with the learning community [56].

## Enhancing students' vocabulary and vocabulary learning motivation

According to [65], vocabulary is the entire set of words that make up the basic linguistic building blocks used to evaluate students' understanding. Not being able to find the words we need to express ourselves is the most frustrating experience when speaking a second language [66]. However, in the traditional English vocabulary learning process, students only memorize words by means of boring recitation and repetitive practice. This phenomenon greatly reduces their curiosity and sense of novelty, making their initial learning interest and enthusiasm disappear [67,68]. When students fail to incorporate effective learning strategies or are not engaged enough to acquire vocabulary, they often give up learning the language [69]. This phenomenon is prevalent among Chinese EFL secondary vocational students [70]. Traditionally, teachers in Chinese EFL vocational situations frequently depend on textbook materials to improve students' language proficiency. They prioritize language activities that involve exercises and emphasize grammatical practice. Students have to rely on extensive paper-based vocabulary exercises, a method that often reduces engagement and limits contextualized learning opportunities. This approach leaves little room for vocabulary development, resulting in students' demotivation in learning [71]. Recent study [72] underscores the need for adaptive and interactive assessment modalities to enhance both vocabulary acquisition and learner motivation. Besides, providing a stimulating English learning atmosphere to boost their vocabulary learning motivation for this particular cohort of students is imperative [49].

The growth of digital technology has made it possible to contextualize foreign language learning in real-world environments [49]. Vocabulary has been one of the most commonly emphasized aspects of language acquisition through blended learning approaches [73]. This approach offers students an entirely novel educational experience. By interacting with digital platforms, they may access more materials and use multimedia such as texts, photos, audio, and videos to meet different learning styles [74]. These visual aids facilitate students' comprehension of language [75] and enhance their learning outcomes and motivation [76–79]. Wang and Chen [73] assert that a collaborative blended learning environment is more effective for vocabulary acquisition and has shown useful for vocabulary growth. In a collaborative blended learning setting, it is anticipated that students' motivation, emotions, cognition, and metacognitive abilities can be strengthened [80]. Furthermore, students are noted to actively participate in knowledge production discourse as community members, ultimately improving their vocabulary performance [81].

While research on the efficacy of Community of Inquiry (CoI) frameworks for vocabulary development is sparse, various studies underscore its beneficial influence on English learning outcomes and motivation in blended learning environments [55,82,83]. Hasanah and Surya [84] discovered that a CoI-based blended learning strategy helps improve students' performance in an online English course with their grammar, accuracy, and vocabulary enhanced; Herrera and González [16] demonstrated that TP enhances oral competencies, encompassing vocabulary and grammar, whereas Mo and Lee [85] illustrated its effectiveness in improving L2 competency and confidence. Research indicates significant relationships

among the three presences of the CoI (Teaching Presence (TP), Social Presence (SP), Cognitive Presence (CP)), perceived learning, and engagement, with TP and CP being especially predictive of English learning outcomes [9,86]. Moreover, teachers' prompt feedback and facilitated discourse, markedly enhances EFL students' online engagement [15]. These studies collectively highlight the effectiveness of CoI framework in blended EFL environments for improving both students' performance and engagement.

To design of effective blended and online courses, educators are trying to integrate different digital tools for language classes. Digital tools provide English language learners the chance to practice their language skills continuously, receive immediate feedback, and assess their progress in real time, all of which are essential for improving their language retention and skill set [87]. In order to monitor students' progress in real time, several nations have long integrated digital tools including Google Classroom, Edmodo, Socrative, Classting, Madrasati, Rosetta Stone, and Busuu [88]. Teachers can use these tools to create automatically graded projects and quizzes that give instant feedback [89]. In China, Chaoxing Learning App is a professional mobile learning platform that offers portability, uniqueness, accessibility, adaptability, persistence, utility, and usability, making it an ideal tool for language learning. To achieve the learning objectives of this study, it was selected as an online learning platform to complement traditional face-to-face instruction.

## Methodology

An explanatory sequential mixed methods approach was used in this study. According to [90], an explanatory sequential mixed-method study typically consists of quantitative data collection in the first phase, followed by qualitative data collection in the second phase. In this study, quantitative data was firstly collected via a quasi-experiment, while qualitative data was then collected through a semi-structured interview.

### Participants

According to [91], in quasi-experimental research, respondents cannot be randomly assigned to conditions. Under this premise, a total of 86 first-year students aged 14–17 from two natural classes of a Technician College in Chongqing, China, were selected as the participants of this study. They were assigned to an experimental group (n = 44) and a control group (n = 42) for comparison. Using purposive sampling, both groups were matched based on baseline vocabulary scores and demographics, with pre-test homogeneity confirmed (t (84) = 0.20, p = 0.83). After the CoI-based blended learning intervention, 12 students from the experimental group were interviewed on a voluntary basis.

### Instruments

**Vocabulary pre-test and post-test.** Multiple-choice assessment is the most often employed evaluations to measure the receptive vocabulary knowledge of EFL/ESL learners [92]. In this study, a researcher-developed Vocabulary Test was employed for both the pre-test and post-test. The test consisted of 40 multiple-choice questions, each valued at one point, yielding a maximum score of 40. Students were mandated to select one accurate response from the four alternatives: A, B, C, or D. The Vocabulary Test questions were subsequently prepared in accordance with the national standard English textbook for secondary vocational students: *New Model English 1 (2nd edition)*. The items were established in accordance with the standard Classified College Entrance Examination (CCEE) test package. Following the creation of the draft, it was subjected to evaluation and approval by a panel of two experts in Chinese secondary vocational EFL education. The Cronbach's Alpha for the Vocabulary Test was 0.718, indicating an acceptable level [93].

Prior to the CoI-based blended learning intervention, a pre-test was administered using the aforementioned researcher-developed Vocabulary Test to assess the pre-existing Vocabulary Test Scores (VTS) of the two target groups. The objective was to evaluate the students' initial proficiency and set a baseline for comparison in the post-test. The post-test is a standardized evaluation administered at the conclusion of the intervention. It seeks to evaluate students

attainment of the particular intervention [94]. Naps et al. [95] asserted that care must be given to ensure that the pre-test and post-test are isomorphic. This is most effectively achieved by employing the same questions in a varied sequence. Consequently, the same Vocabulary Test utilized in the pre-test was administered in the post-test, with the items randomized and reorganized to prevent students from depending on rote memorization. To maintain validity, the students were not informed of the post-test, preventing them from recalling the words or questions from the pre-test [96].

**Vocabulary learning motivation questionnaire.** The VLM questionnaire consisted of two sections. The first section aimed to gather demographic information from students, including gender and fundamental details regarding their English language acquisition. The second section concentrated on the VLM inquiries. This questionnaire was adapted from [56], who cross-culturally modified the Vocabulary Learning Motivation Questionnaire [97] and validated it within a Chinese high school context. Initially comprising 20 items, it was subsequently refined to 19 items following the two aforementioned experts evaluation due to redundancy in the descriptions of two questions. Students were instructed to respond to all inquiries within each section using a five-point Likert-type scale, ranging from "strongly agree" (five points) to "strongly disagree" (one point). The Cronbach's Alpha for the VLM questionnaire indicated a reliability of 0.855, categorized as a Good level [93]. Prior to the CoI-based blended learning intervention, the pre-intervention was conducted among the students in both groups using this VLM questionnaire. In the post-intervention phase, the same VLM questionnaire was administered but with the items rearranged and reordered.

**Semi-structured interview protocol.** After the CoI-based blended learning intervention, semi-structured interviews were performed with 12 students from the experimental group. The interviews included 15 open-ended questions, and participation was voluntary. The objective was to investigate students' perspectives of the CoI-based blended learning approach and to elucidate the quantitative findings further. Initially, 10 open-ended questions were formulated. Given the traits of secondary vocational school students, who typically exhibit reticence, two local experts proposed a final version of 15 open-ended questions.

## Data collecting and analysis

In the first phase, the researcher obtained ethical approval from the Technician College. Prior to the study, all participants received comprehensive information regarding the objectives and methodologies of this study. They were obligated to provide written consent forms for participation. Prior to the main study, a pilot study was conducted to obtain feedback regarding the reliability of the instruments [98].

In the second phase, the researcher started the study on September 1, 2022. In the first week of the first semester of the 2022–2023 academic year, the researcher administered the Vocabulary Test and VLM questionnaire for both the experimental and control groups to gather pre-test and pre-intervention data. The data were gathered using Questionnaire Star, a prominent survey tool in China. Subsequently, starting in the second week, the experimental group initiated the CoI-based blended learning intervention for vocabulary acquisition. Conversely, the control group continued to use the traditional face-to-face approach. Falleti et al. [99] proposed that individuals should be assessed at intervals of no less than one month to accurately reflect their cognitive skills. Therefore, this study was conducted over a duration of 13 weeks to mitigate the potential for practice effects. Subsequently, the post-test and post-intervention data were collected using Questionnaire Star the week following the completion of the 13-week CoI-based blended learning intervention. Students in both groups were administered the same Vocabulary Test and VLM questionnaire, but with the items disordered and rearranged. In the week subsequent to the post-test and post-intervention, 12 students from the experimental group were voluntarily participated for semi-structured interviews. To guarantee clarity, the researcher orally delivered all questions in a bilingual style to the students, and the respondents' opinions and comments answered in Chinese were recorded according to their agreements. The interviews were conducted on November 30, 2022, signifying the completion of this study.

During the third phase, the quantitative data were analyzed using SPSS version 26.0. Descriptive analysis was conducted to reveal the demographic information and students' VTS and VLM scores. The Shapiro-Wilk (S-W) test was employed to assess the normality of the data distribution. To assess students' achievement of the intervention, paired samples t-test and independent samples t-test were applied. Cohen's d and the Confidence Interval were used to compute the effect size. The formula for Cohen's d is:

$$d = \frac{M_1 - M_2}{\sqrt{\frac{(SD_1^2 + SD_2^2)}{2}}}$$

where: 1) $M_1$ and $M_2$ are the means of the two groups being compared. 2) $SD_1$ and $SD_2$ are the standard deviations of the two groups. The standard error of Cohen's d was used to calculate the confidence intervals. The formula for the standard error is:

$$SE_d = \sqrt{\frac{n_1 + n_2}{n_1 n_2} + \frac{d^2}{2(n_1 + n_2)}}$$

where: $n_1$ and $n_2$ are the sample sizes of the two groups. The 95% confidence intervals for Cohen's d were calculated using the standard error. The formula for the confidence intervals is:

$$CI = d \pm 1.96 \times SE_d$$

where: 1) $d$ is the calculated Cohen's $d$. 2) SE$d$ is the standard error of Cohen's $d$. 3) 1.96 is the z-score corresponding to a 95% confidence level (for a two-tailed test). Conversely, qualitative data were analyzed via thematic analysis. According to [100], to achieve trustworthiness, selecting individuals with diverse backgrounds enhances the probability of providing insight into the study subject from multiple perspectives. Therefore, the interviewees were students of four high proficiency level, four medium proficiency level, and four low proficiency level students according to their Vocabulary Test scores. Initially, written transcripts were generated from the interview recordings. The transcripts were thereafter printed and provided to the interviewers for verification of their alignment with the original meaning. Subsequently, the Chinese interview data was translated into English and submitted to a translator for verification. Inter-coder reliability assessments have been suggested to enhance reliability by evaluating concordance among several coders [101]. According to Miles and Huberman [102]'s guidelines for two researchers working together on qualitative analysis. Therefore, thematic analysis was completed by the researcher and teacher A (a lecturer in TESOL with more than ten years of teaching experience). The coding process follows the Open, Axial, and Selective Coding approaches by [103]. Codes were generated to satisfactory levels after the researcher and teacher A's discussions and clarifications. Upon the conclusion of the three procedures, two Chinese experts in Chinese vocational EFL teaching field reviewed the coding process and provided suggestions concerning the emerging themes and sub-themes. A sample of the coding is shown in Table 1.

### Intervention plan

According to the timetable of the Technician College, the 13-week quasi-experiment consisted of 40 class sessions, each lasting 40 minutes and occurring four times a week. The instructional material was the national standard English textbook for secondary vocational students, titled *New Model English 1 (2nd edition)*. The Chaoxing Learning App, a technology-based implementation tool that ensures classroom and online interaction, was selected as a complementary online learning platform to traditional face-to-face instruction. The experimental group relied on it to preview and review, complete activities, assignments, and quizzes, as well as to engage in discussions both in and out of class. Comparatively, the control group received instruction on the targeted vocabulary through face-to-face approach.

**Table 1. Coding example of interview transcripts.**

| Questions & Answers | Open coding | Axial coding | Selective coding (Sub-themes) | Themes |
|---|---|---|---|---|
| S1-Q3: Did you feel motivated to learn vocabulary in this CoI-based blended learning course? A: *I think my motivation for learning is pretty good because, in this way, we can be more engaged. Then sometimes the teacher uses some encouragement methods. For example, every week the teacher evaluates the top three students in the system and rewards them. This makes us more motivated to learn. In addition, I think the Chaoxing Learning app is very practical and useful, there are so many functions and these functions brought us fun activities.* | 1) The teacher uses some encouragement methods which made the students more motivated to learn; 2) The Chaoxing Learning app is very practical and useful with many functions. 3) Chaoxing Learning app provides the students fun activities. | 1) Teachers take encouraging measures; 2) The platform has practical and useful functions; 3) Fun activities taken to motivate students; | 1) Confidence encouraging; 2) Useful functions; 3) Interest stimulating; | 1) Emotion; 2) Communication method; |
| S2-Q4: What do you think the strengths and weaknesses of vocabulary acquisition are in this CoI-based blended learning course? A: *The disadvantage is that secondary school students are more or less unconscious, although they are usually very active in the classroom, there are inevitably a few students who take the opportunity to browse other websites when having class. Another disadvantage is that we have to do the homework on the mobile phone, and some of the answers can be found directly online. Some students will just copy and paste it over without using their brains at all. The advantage is that it can improve the enthusiasm of students because mobile phones have a youthful nature. For students of our age, being able to get along with mobile phones will make us very excited, and then those students who are sleeping can be effectively stopped.* | 1) A few students browsed other websites during the class; 2) Some students just copy and paste the answers online; 3) This CoI-based blended learning approach improved students' enthusiasm; 4) Mobile phones made students excited. | 1) Students autonomous learning is not good; 2) Students self-control ability should be enhanced; | 1) Autonomous learning; 2) Self-control ability; | 1) Supervision; |
| S3-Q5: What do you think of the online and offline discussion in this CoI-based blended learning environment? A:*I feel that in this blended learning environment, there is much more interaction between the teacher and the students and between the students themselves than in the traditional classroom. This mutual interaction is very helpful to enhance the relationship between the students and also between the teacher and the students. Students often discuss and solve problems proactively and enthusiastically online and offline together. They do not evade responsibility. Everyone participates in discussions actively, voluntarily, and proactively.* | 1) There is much more interaction between the teacher and the students and between the students themselves in this CoI-based blended learning environment; 2) Mutual interaction is very helpful in enhancing the relationship between the students and also between the teacher and the students; 3) Students often discuss and solve problems proactively and enthusiastically online and offline together. | 1) Interactions occurred among the teacher and students; 2) The relationship between the teacher and students was good; 3) Students solve problems actively. | 1) Effective communications; 2) Teacher-students, Students-students; 3) Solutions digging. | 1) Interaction; 2) Affective collection; 3) Meaningful learning |

Following the introduction of the course orientation and the blended learning model in the first week. Students in the experimental group (n = 44) were divided into seven communities. Each community was associated with a corresponding online learning community on the Chaoxing Learning App. Students were designated diverse roles including group leader, technical recorder, designer, and connector. Prior to class, students were required to finish both individual and group work. They were encouraged to raise any confusion related to specific course topics. Using the Chaoxing Learning App with blended learning terminals, they were required to submit class-related information they had collected and present their

inquiries. During the class, volunteers and randomly chosen students presented the assigned tasks during the face-to-face meeting. The instructor offered comprehensive feedback and structured knowledge regarding their presentation. Following the presentations, peers were obligated to share their reflections and insights on the discussion forum. A random roll call, quizzes, and on-screen remarks were conducted using the Chaoxing Learning App. After class, students engaged in knowledge integration, completed individual and group activities, and ultimately their confusions and problems resolved.

In the CoI-based blended learning environment, students with superior English proficiency served as scaffolds to assist other group members, fostering a learning community centered on inquiry. Students inside a learning community were encouraged to participate in both individual and collaborative learning while interacting with their teacher and peers. To enhance deep learning and higher-order thinking skills, various activities were implemented during the teaching and learning process by employing the three elements of the CoI framework. Table 2 delineates the sample instructional plan.

## Results

A total of 86 first-year students (2022–2023 academic year) belonging to two classes from a Technician College in Chongqing, China, were involved in this study. Their age ranged from 14–17. 44 students of them were assigned to the experimental group, while the other 42 students were allocated to the control group for comparison. The experimental group had 81.8 percent male students (n = 36) and 18.2 percent female students (n = 8), while the control group consisted of 85.7 percent male students (n = 36) and 14.3 percent female students (n = 6) (see Table 3).

Table 4 shows the descriptive analysis of students' VTS and VLM pre-test/ pre-intervention and post-test/ post-intervention scores of both the experimental and control groups. It is clear to know that the mean scores of the experimental group are greater than those of the control group in the post-test/ post-intervention scores with regard to their VTS and VLM after 13 weeks of CoI-based blended learning intervention.

Thode [104] claims that the Shapiro-Wilk test is the optimal method for assessing normality in data when sample sizes are below 50. Upon analyzing the p-values in the Sig. column of the Shapiro-Wilk tests presented in Table 5, it is evident that all $p$-values exceeded 0.05. Therefore, the distributions of the Vocabulary pre-test and post-test scores, as well as the VLM pre-intervention and post-intervention scores for both the experimental and control groups, were normally distributed. The parametric paired samples t-test and independent samples t-test can be proceeded.

What is the significant difference of using CoI-based blended learning and face-to-face approaches intervention in: 1A1: students VTS in terms of their pre-test and post-test scores ?

Based on Levene's test, Table 6 shows equal variance with $F = 0.01$ and $p = .910$. It was illustrated that there was no statistically significant difference in Vocabulary pre-test scores between the experimental group ($M = 15.64$, $SD = 6.36$) and the control group ($M = 15.60$, $SD = 5.99$) with $t$ (84) = 0.20, $p = .830$. Based on their Vocabulary pre-test, the research sample groups were homogeneous in regard to their VTS.

According to Levene's test presented in Table 7, equal variances can be assumed ($F = 0.48$, $p = .480$). According to the results of the independent t-test, students in the experimental group ($M = 25.43$, $SD = 7.96$) performed significantly better after the intervention than those in the control group ($M = 17.05$, $SD = 7.43$) with $t$ (84) = −5.04, $p < .001$. According to [105], Cohen's d values of 0.2, 0.5, and 0.8 are typically considered to represent small, medium, and large effects, respectively. The Cohen's d effect size for this comparison was 1.09, with a 95% CI [0.92, 1.25], which indicates a large effect size, suggesting that it is 95% confident that the true effect size lies within this range. The wide interval, which does not include zero, further supports the conclusion that the CoI-based blended learning approach significantly enhanced students' VTS compared to the traditional face-to-face method.

A paired-sample t-test was conducted to compare the Vocabulary pre-test and post-test scores of the control group students. Table 8 demonstrates that there was no significant difference between their Vocabulary pre-test ($M = 15.60$,

**Table 2. The Application CoI Framework- Take lesson *let's go shopping* as an example.**

| Teaching Steps | Teachers Activities | CoI Application | Students Activities | CoI Application |
|---|---|---|---|---|
| **Before- Class** | **Preview (Online)** 1. Create an online course and groups with the ChaoXing Learning App, and instruct students to log in using a QR code. | **Teaching Presence (TP)-** Design & Organization | **Preview (Online)** 1. Sign in the class and belonged group according to the invitation QR code. | -- |
| | **Preview (Online)** 2. Release a short video -"shopping" to students on the ChaoXing Learning App: https://m.v.qq.com/play.htm-l?vid=h07651mgg1c&cid =, and ask them to finish preview task card 1 by group. | **Teaching Presence (TP)-** Design & Organization; Facilitating Discourse **Social Presence (SP) -** Affective Expression; Group Cohesion | **Preview (Online)** 2. Watch the short video-"shopping" and finish preview task card 1 by group. | **Cognitive Presence (CP)-** Triggering Event **Social Presence (SP) -** Affective Expression; Group Cohesion |
| | **Preview (Online)** 3. Instruct students to read and preview the short reading in Exercise A on page P21, the important phrases in Exercise F on page P22, and the ten key words on page P164, and to complete preview task card 2 individually. | **Teaching Presence (TP)-** Design & Organization | **Preview (Online)** 3. Complete the preview tasks of the short reading in Exercise A on page P21, Key sentences in Exercise F on page P22, and 10 key words on page P164 and finish pre-view task card 2 personally. | **Cognitive Presence (CP)-** Triggering Event |
| | **Preview (Online)** 4. Instruct students to research Martins Department Store, 24−7 Convenience Store, Sams Food Market, Hero Books, Shoe Empo-rium, Dress for Less, and Thrift Central, and com-plete preview task card 3 collaboratively. | **Teaching Presence (TP)-** Design & Organization; Facilitating Discourse **Social Presence (SP) -** Affective Expression; Open Communication; Group Cohesion | **Preview (Online)** 4. Investigate details regard-ing Martins Department Store, 24−7 Convenience Store, Sams Food Market, Hero Books, Shoe Emporium, Dress for Less, and Thrift Central, then complete the preview task card 3 collabora-tively as a group. | **Cognitive Presence (CP)-** Triggering Event **Social Presence (SP) -** Affective Expression; Open Communication; Group Cohesion |
| | **Preview (Online)** 5. Release a micro lesson on the use of lexical verbs in the simple present tense: https://b23.tv/0oOhiL6 Instruct students on the ChaoXing Learning App to collabora-tively complete preview task card 4. | **Teaching Presence (TP)-** Design & Organization; Facilitating Discourse **Social Presence (SP) -** Affective Expression; Open Communication; Group Cohesion | **Preview (Online)** 5. Watch the micro lesson of lexical verb used in Simple present tense: https://b23.tv/0oOhiL6, and finish preview task card 4 by group. | **Cognitive Presence (CP)-** Triggering Event **Social Presence (SP) -** Affective Expression; Open Communication; Group Cohesion |
| | **Preview (Online)** 6. Ask students to upload preview tasks cards to the ChaoXing Learning App and do the self-assessment of preview tasks. | **Teaching Presence (TP)-** Design & Organization | **Preview (Online)** 6. Finish preview task cards and upload them to the ChaoXing Learning App and do the self-assessment of preview tasks. | -- |
| **In-Class** | **Lead-in (Online+Face-to-face)** 1. Issues a sign-in gesture on the ChaoXing Learning App. **(1 min)** | **Teaching Presence (TP)-** Design & Organization | **Lead-in (Online+Face-to-face)** 1. Sign-in through the ChaoX-ing Learning App by gesture. | -- |

*(Continued)*

| Teaching Steps | Teachers Activities | Col Application | Students Activities | Col Application |
|---|---|---|---|---|
| | **Lead-in (Online+Face-to-face)** 2. Present students with images related to shopping and pose the question: Do you enjoy shopping? Request students to generate and list the vocabulary related to shopping on the screen. Present the subject of the current lesson. **(2 min)** | **Teaching Presence (TP)-** Design & Organization; Facilitating Discourse **Social Presence (SP) -** Affective Expression; Open Communication | **Lead-in (Online+Face-to-face)** 2. Respond to the instructor's inquiries. Generating and listing the vocabulary related to shopping that they are familiar with on the screen. | **Cognitive Presence (CP)-** Triggering Event **Social Presence (SP) -** Affective Expression; Open Communication |
| | **Vocabulary Learning (Online+Face-to-face)** 3. Read about Van on page 21 with students and introduce Van's problem to students. **(2 min)** | **Teaching Presence (TP)-** Design & Organization; Facilitating Discourse **Social Presence (SP) -** Affective Expression; Open Communication | **Vocabulary Learning (Online+Face-to-face)** 3. Read about Van on page 21with the teacher and get to know Van's problem | **Cognitive Presence (CP)-** Triggering Event **Social Presence (SP) -** Affective Expression; Open Communication |
| | **Vocabulary Learning (Online+Face-to-face)** 4. Explain 10 key words and 1 sentence pattern of this lesson. **(8 min)** | **Teaching Presence (TP)-** Design & Organization; Facilitating Discourse; Direct Instruction **Social Presence (SP) -** Affective Expression; Open Communication. | **Vocabulary Learning (Online+Face-to-face)** 4. Learn 10 key words and 1 sentence pattern of this lesson. | **Cognitive Presence (CP)-** Exploration **Social Presence (SP) -** Affective Expression; Open Communication. |
| | **Vocabulary Learning (Online+Face-to-face)** 5. Release the Seewo words game through the ChaoXing Learning App and announce the rules to the students. **(8 min)** | **Teaching Presence (TP)-** Design & Organization; Facilitating Discourse **Social Presence (SP)-** Open Communication; Group Cohesion | **Vocabulary Learning (Online+Face-to-face)** 5. Each group selects one representative to compete in pairs, with the victor remaining until the conclusion of the game. | **Cognitive Presence (CP)-** Exploration **Social Presence (SP)-** Open Communication; Group Cohesion |
| | **Vocabulary Learning (Online+Face-to-face)** 6. Select students at random to present the information they gathered on various stores before class. **(6 min)** | **Teaching Presence (TP)-** Design & Organization; Facilitating Discourse **Social Presence (SP) -** Affective Expression; Open Communication; Group Cohesion | **Vocabulary Learning (Online+Face-to-face)** 6. Present the information about different stores that they collected before class by group. | **Cognitive Presence (CP)-** Exploration **Social Presence (SP) -** Affective Expression; Open Communication; Group Cohesion |
| | **Vocabulary Learning (Online+Face-to-face)** 7. Ask students to help Van find the right stores to buy her needed items and finish Exercise B & C. **(5 min)** | **Teaching Presence (TP)-** Design & Organization; Facilitating Discourse; Direct Instruction **Social Presence (SP) -** Affective Expression; Open Communication | **Vocabulary Learning (Online+Face-to-face)** 7. Help Van find the right stores to buy her needed items and finish Exercise B & C. | **Cognitive Presence (CP)-** Exploration **Social Presence (SP) -** Affective Expression; Open Communication |
| | **Vocabulary Learning (Online+Face-to-face)** 8. Provide directives for students to conduct Exercise D: Listen to Van and her spouse. Identify and circle the best location to get each item. **(3 min)** | **Teaching Presence (TP)-** Design & Organization; Facilitating Discourse; Direct Instruction **Social Presence (SP) -** Affective Expression; Open Communication. | **Vocabulary Learning (Online+Face-to-face)** 8. Complete Exercise D: Listen to Van and her husband. Circle the best place to get each item. | **Cognitive Presence (CP)-** Exploration **Social Presence (SP) -** Affective Expression; Open Communication. |

*(Continued)*

| Teaching Steps | Teachers Activities | CoI Application | Students Activities | CoI Application |
|---|---|---|---|---|
| | **Conversation Practicing (Online+Face-to-face)** 9. Instruct students to utilize the Youdao Learning App to capture images of the sentence pattern from Exercise F on page 22 and subsequently read aloud using the App. **(5 min)** | **Teaching Presence (TP)-** Design & Organization; Facilitating Discourse | **Conversation Practicing (Online+Face-to-face)** 9. Utilize the Youdao Learning App to capture images of the phrase structure from Exercise F on page 22 and subsequently read them using the App. | **Cognitive Presence (CP)-** Exploration |
| | **Conversation Practicing (Online+Face-to-face)** 10. Students are randomly chosen via the "shake" function of the ChaoXing Learning App to construct phrases incorporating "shoes", "shirts", "dictionary", "bread", "cheese", and "fruit" within new dialogues utilizing the acquired sentence structure. **(7 min)** | **Teaching Presence (TP)-** Design & Organization; Facilitating Discourse **Social Presence (SP) -** Affective Expression; Open Communication | **Conversation Practicing (Online+Face-to-face)** 10. The chosen students construct phrases using "shoes," "shirts," "dictionary," "bread," "cheese," and "fruit" within new dialogues utilizing the learnt sentence structure. | **Cognitive Presence (CP)-** Exploration **Social Presence (SP) -** Affective Expression; Open Communication |
| | **Grammar Tips (Online+Face-to-face)** 11. Facilitate a theme discourse on the ChaoXing Learning App and instruct students to analyze, deliberate, and synthesize the use of the term "shop" in various phrases within Exercise G: Examine the chart alongside your classmates and instructor. **(5 min)** | **Teaching Presence (TP)-** Design & Organization; Facilitating Discourse **Social Presence (SP) -** Affective Expression; Open Communication | **Grammar Tips (Online+Face-to-face)** 11. Refer to Exercise G: Engage in a debate regarding the teacher's inquiry and submit your response on the ChaoXing Learning App via theme discourse. | **Cognitive Presence (CP)-** Exploration **Social Presence (SP) -** Affective Expression; Open Communication |
| | **Grammar Tips (Online+Face-to-face)** 12. Provide more instances of lexical verbs utilized in the simple present tense and instruct students to summarize and conclude. Exercise H: Fill in the sentences with the appropriate form of the verb "shop". **(6 min)** | **Teaching Presence (TP)-** Design & Organization; Direct Instruction | **Grammar Tips (Online+Face-to-face)** 12. Summarize the application of lexical verbs in the simple present tense and conclude. Exercise H: Fill in the sentences with the appropriate form of the verb "shop". | **Cognitive Presence (CP)-** Exploration |
| | **Summary (Online+Face-to-face)** 13. Distribute an online survey via the ChaoXing Learning App and request students to create a bar graph to analyze the frequency of classmates purchasing at various types of Chinese retailers by group. **(8 min)** | **Teaching Presence (TP)-** Design & Organization; Facilitating Discourse **Social Presence (SP) -** Affective Expression; Open Communication; Group Cohesion | **Summary (Online+Face-to-face)** 13. Complete the online survey and create a bar graph to analyze the frequency of classmates shopping at various types of Chinese retailers by group. | **Cognitive Presence (CP)-** Integration **Social Presence (SP) -** Affective Expression; Open Communication; Group Cohesion |

*(Continued)*

**Table 2.** (Continued)

| Teaching Steps | Teachers Activities | CoI Application | Students Activities | CoI Application |
|---|---|---|---|---|
| | **Summary (Online+Face-to-face)** 14. Review, summarize and fill in the mind map with knowledge learned in this lesson together with students. **(7 min)** | **Teaching Presence (TP)-** Design & Organization; Facilitating Discourse; Direct Instruction **Social Presence (SP) -** Affective Expression; Open Communication | **Summary (Online+Face-to-face)** 14. Review, summarize and fill in the mind map with knowledge learned in this lesson together with the teacher. | **Cognitive Presence (CP)-** Integration **Social Presence (SP) -** Affective Expression; Open Communication |
| | **Evaluation (Online+Face-to-face)** 15. Facilitate students in completing the self-assessment form, peer evaluation form, and teacher evaluation form via the ChaoXing Learning App. **(5 min)** | **Teaching Presence (TP)-** Design & Organization; Facilitating Discourse; Direct Instruction **Social Presence (SP) -** Affective Expression; Open Communication; Group Cohesion | **Evaluation (Online+Face-to-face)** 15. Complete the students self-evaluation form, peer evaluation form through the ChaoXing Learning App. | **Cognitive Presence (CP)-** Integration **Social Presence (SP) -** Affective Expression; Open Communication; Group Cohesion |
| After- class | **Expanding (Online)** 1. Discharge Active Task. Visit a shopping mall or utilize the Internet. Identify the names of three apparel retailers that you favor. Prepare a report and post it to the ChaoXing Learning App as a group. | **Teaching Presence (TP)-** Design & Organization; Facilitating Discourse; Affective Expression **Social Presence (SP)-** Open Communication; Group Cohesion | **Expanding (Online)** 1. Execute Active Task. Visit a shopping mall or utilize the Internet. Identify the names of three apparel retailers that you favor. Prepare a report and post it to the ChaoXing Learning App as a group. | **Cognitive Presence (CP)-** Resolution **Social Presence (SP) -** Affective Expression; Open Communication; Group Cohesion |
| | **Expanding (Online)** 2. Instruct students to complete pertinent exercises in their exercise books and upload photographs of them to the ChaoXing Learning App. | **Teaching Presence (TP)-** Design & Organization | **Expanding (Online)** 2. Finish relevant exercises on exercise book and upload it to the ChaoXing Learning App by photos. | **Cognitive Presence (CP)-** Resolution |
| | **Expanding (Online)** 3. Ask students to finish vocabulary exercise on the ChaoXing Learning App. | **Teaching Presence (TP)-** Design & Organization | **Expanding (Online)** 3. Finish vocabulary exercise on the ChaoXing Learning App. | **Cognitive Presence (CP)-** Resolution |
| | **Evaluation (Online)** 4. Assess students' homework and prompt them to conduct self-evaluations of their extra-curricular assignments. | **Teaching Presence (TP)-** Design & Organization | **Evaluation (Online)** 4. Do self-assessment about their after-class tasks. | -- |

$SD = 5.99$) and post-test ($M = 17.05$, $SD = 7.43$) with $t (41) = −0.993$, $p = .326$. It indicates that the conventional face-to-face approach has no statistically significant impact on control group students' VTS.

According to Table 9, a significant difference was found between students Vocabulary pre-test scores ($M = 15.64$, $SD = 6.36$) and post-test scores ($M = 25.43$, $SD = 7.96$) in the experimental group with $t (43) = −8.627$, $p < .001$. The Cohen's d effect size for this improvement was 1.36, with a 95% CI [0.89, 1.82], indicating a large effect size and suggesting that it is 95% confident that the true effect size lies within this range. This substantial increase in VTS suggests that the CoI-based blended learning approach had a significant positive impact on students'vocabulary acquisition.

**Table 3. Demographic characteristics by group.**

| Group | Students (N) | Grade | Age range | Gender | | | |
|---|---|---|---|---|---|---|---|
| | | | | M | P (%) | F | P (%) |
| Experimental | 44 | First-year | 14-17 | 36 | 81.8 | 8 | 18.2 |
| Control | 42 | First-year | 14-17 | 36 | 85.7 | 6 | 14.3 |

*M: Male; F: Female; P: Percentage.

**Table 4. Descriptive statistics of VTS and VLM by group.**

| Group | Items | Test | N | Min | Max | M | SD |
|---|---|---|---|---|---|---|---|
| Experimental group | VTS | Pre-test | 44 | 5 | 30 | 15.64 | 6.36 |
| | | Post-test | 44 | 12 | 40 | 25.43 | 7.96 |
| | VLM | Pre-intervention | 44 | 2.47 | 4.02 | 3.35 | 0.39 |
| | | Post-intervention | 44 | 2.53 | 4.85 | 3.73 | 0.60 |
| Control group | VTS | Pre-test | 42 | 3 | 30 | 15.60 | 5.99 |
| | | Post-test | 42 | 5 | 34 | 17.05 | 7.43 |
| | VLM | Pre-intervention | 42 | 1.73 | 4.75 | 3.20 | 0.69 |
| | | Post-intervention | 42 | 2.05 | 4.47 | 3.33 | 0.60 |

*VTS: Vocabulary test scores; VLM: Vocabulary learning motivation ; N: Number; Min:Minimum ; Mam: maximum; M:Mean; SD:standard deviation.

**Table 5. Normality test results for VTS and VLM measures by group.**

| Group | Item | Shapiro-Wilk | | |
|---|---|---|---|---|
| | | Statistic | df | Sig. |
| **Experimental group** | VTS-pre | .951 | 44 | .059 |
| | VTS-post | .949 | 44 | .052 |
| | VLM-pre | .970 | 44 | .297 |
| | VLM-post | .958 | 44 | .111 |
| **Control group** | VTS-pre | .952 | 42 | .073 |
| | VTS-post | .955 | 42 | .100 |
| | VLM-pre | .984 | 42 | .818 |
| | VLM-post | .979 | 42 | .641 |

*VTS: Vocabulary test scores; VLM:Vocabulary learning motivation.

**Table 6. Independent samples t-test results for vocabulary pre-test scores between groups.**

| Levenes Test for Equality of Variances | | F | Sig. | t-test for Equality of Means | | | | | 95% CI | |
|---|---|---|---|---|---|---|---|---|---|---|
| | | | | t | df | Sig. (2-tailed) | Mean | SD | Lower | Upper |
| VTS | Equal variances assumed | 0.01 | 0.910 | 0.20 | 83 | 0.830 | 0.27 | 1.31 | −2.33 | 2.87 |
| | Equal variances not assumed | | | 0.20 | 82.99 | 0.830 | 0.27 | 1.31 | −2.33 | 2.87 |

*VTS: Vocabulary test scores.

**Table 7. Independent samples t-test results for vocabulary post-test scores between groups.**

| Levenes Test for Equality of Variances | | F | Sig. | t-test for Equality of Means | | | | | | | |
|---|---|---|---|---|---|---|---|---|---|---|---|
| | | | | t | df | Sig. (2-tailed) | Mean | SD | 95% CI | | |
| | | | | | | | | | Lower | Upper |
| VTS | Equal variances assumed | 0.48 | 0.48 | −5.04 | 84 | 0.000 | −8.38 | 1.66 | −11.69 | −5.07 |
| | Equal variances not assumed | | | −5.04 | 83.95 | 0.000 | −8.38 | 1.66 | −11.68 | −5.08 |

*VTS: Vocabulary test scores.

**Table 8. Paired samples t-test results of vocabulary pre-test and post-test scores in the control group.**

| | Paired Differences | | | | | t | df | Sig. (2-tailed) |
|---|---|---|---|---|---|---|---|---|
| | Mean | Std. Deviation | Std. Error Mean | 95% CI | | | | |
| | | | | Lower | Upper | | | |
| VTS | −1.452 | 9.477 | 1.462 | −4.406 | 1.501 | −0.993 | 41 | 0.326 |

*VTS: Vocabulary test scores.

1A2: students VLM in terms of their pre-intervention and post-intervention scores ?

As indicated by Table 10, a non-equal variance was found by Levine's test ($F=8.579$, $p=.004$). Using the independent samples t-test, it was determined that there was no statistically significant difference between the experimental group ($M=3.35$, $SD=0.39$) and the control group ($M=3.20$, $SD=0.69$) in terms of students VLM as indicated by $t$ (64) = −1.062, $p=.292$. It means prior to receiving the CoI-based blended learning intervention as well as conventional face-to-face instruction, the VLM between the two groups were homogeneous.

The result of the independent samples t-test in Table 11 shows a statistically significant difference with $t$ (84) = −2.886, $p=.005$ between the experimental ($M=3.73$, $SD=0.60$) and the control group ($M=3.33$, $SD=0.60$) in terms of the VLM post-intervention scores. The Cohen's d effect size for this difference was 0.67, with a 95% CI [0.56, 0.77]. This indicates a medium effect, suggesting that it is 95% confident that the true effect size lies within this range. This interval, which excludes zero, provides strong evidence that the CoI-based blended learning approach effectively increased students' VLM compared to the control group.

According to Table 12, there was no significant difference in the pre-intervention ($M=3.20$, $SD=0.69$) and post-intervention ($M=3.33$, $SD=0.60$) scores of VLM for control group students with $t$ (41) = −0.98, $p=.320$. In this regard, the conventional face-to-face approach has been demonstrated to not have a significant impact on students VLM.

Similarly, a paired samples t-test indicated in Table 13 that the mean scores of experimental group students VLM post-intervention ($M=3.73$, $SD=0.60$) were significantly higher than their pre-intervention ($M=3.35$, $SD=0.39$). It can be concluded that the CoI-based blended learning approach significantly improved experimental group students' VLM with $t=−4.083$, $p<.001$. The Cohen's d effect size for this improvement was 0.75, with a 95% CI [0.32, 1.18]. This indicates a medium effect, suggesting that it is 95% confident that the true effect size lies within this range. This result highlights the positive influence of the CoI-based blended learning approach on students VLM.

What are students' perceptions of using CoI-based blended learning approach for enhancing VTS and VLM in the context of Chinese secondary vocational schools EFL education?

**Table 9. Paired samples t-test results of vocabulary pre-test and post-test scores in the experimental group.**

| | Paired Differences | | | | | t | df | Sig. (2-tailed) |
|---|---|---|---|---|---|---|---|---|
| | Mean | Std. Deviation | Std. Error Mean | 95% CI | | | | |
| | | | | Lower | Upper | | | |
| VTS | −9.795 | 7.532 | 1.135 | −12.085 | −7.506 | −8.627 | 43 | 0.000 |

*VTS: Vocabulary test scores.

**Table 10. Independent samples t-test results for VLM pre-intervention scores between groups.**

| Levenes Test for Equality of Variances | | F | Sig. | t-test for Equality of Means | | | | | | | |
|---|---|---|---|---|---|---|---|---|---|---|---|
| | | | | t | df | Sig. (2-tailed) | Mean | SD | 95% CI | | |
| | | | | | | | | | Lower | Upper | |
| VLM | Equal variances assumed | 8.579 | 0.004 | −1.068 | 83 | 0.289 | −0.130 | 0.121 | −0.371 | 0.112 | |
| | Equal variances not assumed | | | −1.062 | 64 | 0.292 | −0.130 | 0.122 | −0.374 | 0.114 | |

*VLM: Vocabulary learning motivation.

**Table 11. Independent samples t-test results for VLM post-intervention scores between groups.**

| Levenes Test for Equality of Variances | | F | Sig. | t-test for Equality of Means | | | | | | |
|---|---|---|---|---|---|---|---|---|---|---|
| | | | | t | df | Sig. (2-tailed) | Mean | SD | 95% CI | |
| | | | | | | | | | Lower | Upper |
| VLM | Equal variances assumed | 0.087 | 0.769 | −2.886 | 84 | 0.005 | −0.379 | 0.131 | −0.639 | −0.118 |
| | Equal variances not assumed | | | −2.886 | 83.801 | 0.005 | −0.379 | 0.131 | −0.639 | −0.118 |

*VLM: Vocabulary learning motivation.

**Table 12. Paired samples t-test results of VLM pre- and post-intervention scores in the control group.**

| | Paired Differences | | | | | t | df | Sig. (2-tailed) |
|---|---|---|---|---|---|---|---|---|
| | Mean | Std. Deviation | Std. Error Mean | 95% CI | | | | |
| | | | | Lower | Upper | | | |
| VLM | −0.13 | 0.86 | 0.13 | −0.40 | 0.13 | −0.98 | 41 | 0.32 |

*VLM: Vocabulary learning motivation.

**Table 13. Paired samples t-test results of VLM pre- and post-intervention scores in the experimental group.**

| | Paired Differences | | | | | t | df | Sig. (2-tailed) |
|---|---|---|---|---|---|---|---|---|
| | Mean | SD | Std. Error Mean | 95% CI | | | | |
| | | | | Lower | Upper | | | |
| VLM | 0.384 | 0.625 | 0.094 | −0.574 | −0.195 | −4.083 | 43 | 0.000 |

*VLM: Vocabulary learning motivation.

As a consequence of the semi-structured interview, which supported the quantitative results, the students expressed favorable comments regarding the vocabulary learning based on the CoI-based blended learning approach. These transcripts have been analyzed in the following manner:

### Related to teaching presence

Three themes and nine sub-themes were revealed according to students perceptions of Teaching Presence (TP) (see Table 14).

**Guidance- A) Professional Instruction, B) Timely Feedback, C) Diverse Evaluations.** The analysis of the semi-structured interviews revealed that all of the participants seem to agree that the teacher provided them with professional knowledge and skills to develop their cognitive, affective, and psychomotor skills. This is reflected by the teachers instructional planning, teaching design, classroom management, and evaluation skills. In addition to that, the teacher also provides immediate, constructive, multiple, and critical feedback on students' in-class participation, correctness of their performances, online and offline learning scoring, and emotions. This kind of feedback could provide students with

**Table 14. Themes related to teaching presence.**

| Teaching Presence | Guidance | A) Professional Instruction | *I could feel that the teacher had prepared her class more carefully in this blended learning environment. The teacher incorporated a lot of **professional knowledge** related to the subject of English. It is not as casual as traditional teaching.* (S5) |
|---|---|---|---|
| | | B) Timely Feedback | *We used to have to wait a long time for the teacher to finish correcting our assignments and papers after exams, but with the blended learning model, we can know where we **went wrong in time** and make targeted corrections.* (S10) |
| | | C) Diverse Evaluations | ***Peer assessment** is one of my favorite classroom activities, and whenever the teacher distributes students work, the classroom can be somewhat active.* (S4) |
| | Emotion | A) Behaviour respecting | *In traditional education, teachers may be reluctant to ask students with low grades to answer questions. But in the blended learning approach, independent of our grades, we can engage in quizzes, brainstorming, topic discussions, group projects, etc. I feel that everyone **received adequate attention**.* (S9) |
| | | B) Confidence encouraging | *In the past, I was afraid of answering the teacher's questions because I was afraid that I would make mistakes, but in this blended learning environment, I am active to challenge myself because it is really **encouraging from the teachers side**.* (S12) |
| | | C) Diverse Interest stimulating | *In this blended teaching mode, everyone is more active and **interested especially in learning vocabulary**, and then, in traditional teaching like in junior middle school, the teacher usually follows the book and speaks in a very rigid way, and then you tend to get distracted in class.* (S3, S4, S6) |
| | Supervision | A) Autonomous learning | *As secondary school students are more or less unconscious, although they are usually very active in the classroom, there are inevitably a few students who take the opportunity to **browse other websites** when having class.* (S2) |
| | | B) Self-control ability | *Sometimes group members would **browse other websites** not related to the class, therefore, it is necessary for us to gain the teachers supervision.* (S8) |
| | | C) Diverse Continuous grit | *I feel that learning English is a long-term process, and because our classmates have all been rather disinterested in the past, I was concerned that they would find it **difficult to continue** once the novelty wore off.* (S5) |

the guidance they need to accomplish their goals, allow them to participate in the classroom activities, enable them to learn more about their talents, and help them keep their self-confidence. They also stated that multiple evaluation forms including self-evaluations, peer evaluations, teacher evaluations, and third-party evaluations were utilized in CoI-based blended learning. These evaluations may help them to master their learning materials and enhance learning motivation.

**Emotion- A) Behaviour respecting, B) Confidence encouraging, C) Diverse Interest stimulating.** Students claimed that they felt appreciated and respected for their behavior in the CoI-based blended learning environment. They were considered the center of learning and teaching, encouraged to resolve problems independently, and take responsibility for their own actions. The teacher enhanced the lesson plans and content by using technology-based digital tools in the classroom, saving time and energy in the teaching process. As a result, more time was available to address the students' feelings and encourage them. The teacher's behavior and group activities contributed to their increased self-confidence. Additionally, students' interest both online and in the face-to-face classroom were stimulated by technology-based teaching strategies. These techniques helped create a stimulating, exciting learning environment both online and in the classroom.

**Supervision- A) Autonomous learning, B) Self-control ability, C) Diverse Continuous grit.** The CoI-based blended learning environment was reported to be problematic for students in terms of autonomy, this may be attributed to a number of psychological factors, including emotional instability and a lack of enduring interest. To promote autonomous learning, teachers should fully give guiding roles by providing supportive feedback, unbiased comments, and constructive views during tasks. In addition, students also had difficulty concentrating in class, refraining from cell phone use and distractions, and engaging in gossip with classmates with a lack of self-control abilities. Therefore, students suggested that the teacher should give more supervision during the learning process in the future. Students believed that there is some correlation between grit and English language proficiency. Perseverance should be established with the teacher's supervision to improve students' English skills. However, secondary vocational students lack the ability to persist persistently in tasks in this study.

### Related to social presence

Three themes and nine sub-themes were revealed according to students' perceptions of Social Presence (SP) (see Table 15).

**Interaction- A) Cooperative Collaboration, B) Healthy Competition, C) Effective Communication.** The CoI-based blended learning approach was considered to provide students with a collaborative learning environment, primarily due to online and offline conversations, interactions, and collaborative course assignments. Additionally, students were provided with opportunities to present their work in class through group discussions or offline activities. Furthermore, competition is considered an effective approach to generate difficulties and enhance performance improvement. Specifically, under this CoI-based blended learning environment, students were randomly selected to participate in competitions such as game competitions and answer races. After completing the task, they could receive awards for success. Due to the variety of teaching strategies and activities, including group work, immediate feedback, and independent learning opportunities, students were able to meet their needs for vocabulary learning. Finally, by encouraging dialogue and discussion through online and offline group discussions, the CoI-based blended learning approach not only facilitated effective communication among students but also created unprecedented interaction opportunities.

**Affective collection- A) Teacher-students, B) Students-students, C) Community-community.** Teachers have greater authority and control over their students, so it is particularly important for teachers to establish a positive relationship with students. In this study, students reported experiencing a positive relationship with their teacher in class, and as a result, they were more likely to comprehend the learning material. This was because they were allowed to freely express their ideas to their teacher, which in turn contributed to a higher level of learning motivation. Furthermore, students stated they were more effective in learning as they felt emotionally connected, a key outcome of the CoI-based

**Table 15. Themes related to social presence.**

| Teaching Presence | Interaction | A) Cooperative Collaboration | I could feel that the teacher had prepared her class more carefully in this blended learning environment. The teacher incorporated a lot of **professional knowledge** related to the subject of English. It is not as casual as traditional teaching. (S5) |
|---|---|---|---|
| | | B) Healthy Competition | We used to have to wait a long time for the teacher to finish correcting our assignments and papers after exams, but with the blended learning model, we can know where we **went wrong in time** and make targeted corrections. (S10) |
| | | C) Effective Communication | **Peer assessment** is one of my favorite classroom activities, and whenever the teacher distributes students work, the classroom can be somewhat active. (S4) |
| | Affective collection | A) Teacher-students | In traditional education, teachers may be reluctant to ask students with low grades to answer questions. But in the blended learning approach, independent of our grades, we can engage in quizzes, brainstorming, topic discussions, group projects, etc. I feel that everyone **received adequate attention**. (S9) |
| | | B) Students-students | In the past, I was afraid of answering the teachers questions because I was afraid that I would make mistakes, but in this blended learning environment, I am active to challenge myself because it is really **encouraging from the teachers side**. (S12) |
| | | C) Community-community | In this blended teaching mode, everyone is more active and **interested especially in learning vocabulary**, and then, in traditional teaching like in junior middle school, the teacher usually follows the book and speaks in a very rigid way, and then you tend to get distracted in class. (S3, S4, S6) |
| | Communication method | A) Diversify tools | I think Chaoxing Learning app is very practical and useful, there are so many functions and these functions brought us fun activities. (S1) |
| | | B) Convenient platform | I think the function of **submitting assignments** and **having tests** in Chaoxing Learning App is very good. (S2) |
| | | C) Useful functions | Hope in the future, more digital tools can be introduced to our English class. (S8) |

blended learning environment fostering positive teacher-student relationships. Lastly, blended learning provides students with a stronger sense of community. Since a sense of community is one of the fundamental factors for meaningful vocabulary learning, this study revealed that blended learning environments effectively facilitated the development of positive relationships within these learning communities.

**Communication method- A) Diversify tools, B) Convenient platform, C) Useful functions.** Compared with traditional face-to-face instruction, students responded significantly better to this more vivid and interactive learning approach. However, they observed that Chaoxing Learning App for delivering CoI-based blended learning is insufficient. Therefore, a number of additional digital tools that are engaging, valuable, user-friendly, and useful for interacting with instructors, peers, and content are needed during future learning processes. In the teaching and learning process, a convenient learning management system provides a straightforward means of integrating multiple technologies into a single environment. However, some students believed that some features of Chaoxing Learning App could be improved to make the platform more convenient. Specifically, it should enable students to not only access and download courseware and instructional materials from anywhere at any time, but also organize their own learning process by offering a variety of learning advantages. Moreover, students indicated that useful functions in an online learning platform were significant

supplementary teaching aids for blended learning. Online and offline discussion and interaction with their peers provided them with the opportunity to exchange vocabulary knowledge.

**Related to cognitive presence**

Three themes and nine sub-themes were revealed according to students' perceptions of Cognitive Presence (CP) (see Table 16).

**Strategy- A) Using Digital Resources, B) Using Novel Tools, C) Effective Instructions.** Students reported that the abundance of online resources makes it easier for them to comprehend relevant knowledge and obtain learning motivation in the CoI-based blended learning environment. When vivid visuals such as graphics, visuals, and cartoons are integrated into their assigned activities, their VTS and VLM enhanced significantly. More importantly, through engaging and dynamic activities, this approach not only supports the retention of knowledge but also sparks students' interest. Specifically, students learned vocabulary through various multimedia formats including audio, video, discussion, practical activities, and other digital tools. Additionally, it was found that digital tools facilitate increased interaction in three key dimensions: between students and content, students and teachers, and among students themselves. As a result, their systematic and critical thinking skills were enhanced through consistent communication among teammates during collaborative projects using digital technologies.

**Course Approach- A) Immense Learning, B) Flipped Classroom, C) Learning Communities.** Several interviewees indicated that they experienced immersive learning by combining elements from the virtual and real worlds in the CoI-based blended learning environment. In particular, a number of technologies, including augmented reality, virtual worlds, and virtual reality systems, have been found to increase the interest and enthusiasm of students as well as their ability to acquire new knowledge and skills, thereby providing a significant learning experience for them. Moreover, it was observed by students that the flipped classroom model also encouraged their independent learning and self-study. Through this approach, students engage in strategic collaboration and in-depth discussions. Specifically, they stated that the flipped classroom was beneficial for improving focus, verifying learning, enabling self-assessment, and empowering them to participate in assessment processes. Furthermore, students' engagement and learning performance increased when they developed a sense of belonging to relevant online learning groups. Although students were divided into different communities, they could nevertheless experience significant learning despite geographical dispersion, asynchronous participation, and limited visual contact.

**Meaningful Learning- A) Curiosity Arousal, B) Solutions Digging, C) Ideas Exploration.** Students considered curiosity to be an intrinsic desire for knowledge that motivates them to participate in a variety of activities and enhances their understandings. However, instructional practices may unintentionally stifle curiosity by emphasizing efficiency, disregarding negative emotions, promoting overconfidence, and employing passive learning strategies. Conversely, the learning process is enhanced when students' curiosity is both awakened and protected. In particular, students emphasized that meaningful learning objectives should focus on assisting them in solving problems, drawing conclusions, collecting solutions, and making correct decisions. This is because effective solutions to problems require higher-order thinking, including critical thinking skills such as analysis, evaluation, and creation. Through engaging in well-grounded, creative, and critical thinking, students conducted in-depth exploration of solutions to problems they encountered, thereby achieving knowledge integration and absorption. Furthermore, students believed that the ability to think creatively lies in their capacity to explore new ideas for problem-solving. As evidenced by their experiences, creative thinking led to the generation of new ideas, shifts in perspectives, and the development of innovative approaches to resolving problems.

## Discussion

This study sought to examine the effect of a Community of Inquiry (CoI)-based blended learning approach on students' Vocabulary Test Scores (VTS) and Vocabulary Learning Motivation (VLM) in Chinese secondary vocational EFL

**Table 16. Themes related to social presence.**

| Teaching Presence | Strategy | A) Using Digital Resources | *I never knew before that there were so many rich **extra-curricular resources** for us secondary school students. In the past, when we were taught traditionally, we only listened to the teacher and if we didn't know what to do, we put it aside, but in the blended teaching mode, the teacher provided us with many learning websites and links. if we don't understand what we know in class, we can consolidate it again and again in online classes until we learn it.* (S12) |
|---|---|---|---|
| | | B) Using Novel Tools | *I love **dubbing** the most. I used to be very shy about speaking English, but through the **funny animations** in dubbing, Ive forgotten about being embarrassed to speak English. We can actually learn it when we speak it.* (S6) |
| | | C) Effective Instructions | *Although the effect of cooperative learning is very good, sometimes we are not very clear about some knowledge points. It will be much better if the teacher **guides** us.* (S7) |
| | Course Approach | A) Immense Learning | *The role-playing in the blended learning approach allows us to **immerse ourselves** in it as if we were the protagonists.* (S6) |
| | | B) Flipped Classroom | *This blended learning mode is helpful in overcoming students' shyness and fear through **individual presentations** and **group presentations**.* (S11) |
| | | C) Learning Communities | *The **learning community** is just like a family, we just discuss and solve the problems together. It is helpful for me to better recognize the knowledge.* (S2) |
| | Meaningful Learning | A) Curiosity Arousal | *In this CoI-based blended learning, the teacher would often set us up with suspense to **stimulate our curiosity**. I think it is very important for us to learn new knowledge.* (S10) |
| | | B) Solutions Digging | *In the process of finding the **solutions to the problems**, I believe I gained knowledge.* (S6) |
| | | C) Ideas Exploration | *When the community work together, some **new ideas** come out.* (S4) |

classrooms. The results demonstrated substantial enhancements in both VTS and VLM for students subjected to the CoI-based blended learning method in contrast to those in the conventional face-to-face learning setting. Furthermore, students exhibited favorable perceptions of this CoI-based blended learning approach. This study advances prior research by demonstrating how certain attributes of Teaching Presence (TP), Social Presence (SP), and Cognitive Presence (CP) affect students' VTS and VLM in vocational blended learning contexts.

The findings from the first sub-research question indicated an approximately 9-point vocabulary gain with a large effect size. Chinese secondary vocational students generally start with vocabulary sizes 800−1,000 words beneath the national secondary education benchmark. Conventional vocabulary teaching methods have faced criticism for their excessive dependence on rote memorization and decontextualized word lists. Therefore, this significant enhancement is especially remarkable within the realm of Chinese secondary vocational education. The success of the VTS intervention can be systematically elucidated using the three components of the CoI framework. The TP component was implemented via meticulously crafted digital scaffolding that tackled essential issues in vocational EFL training. Consistent with previous studies [8,14,103,104], the instructor's well organized lesson plans, incorporating pre-class previews, in-class exercises, and post-class reinforcement, guaranteed that students got scaffolded vocabulary training. The multimodal vocabulary presentation, which integrated video, text, and audio, boosted students' involvement and facilitated students' transition from rote memory to profound cognitive thinking [106]. Furthermore, interviewees indicated that the instructor's feedback markedly enhanced their confidence. This corresponds with [107], indicating that the teacher's role in facilitating discourse,

delivering immediate feedback, and controlling group interactions assisted students in consolidating language. Social Presence was similarly vital to the success of the intervention, especially in tackling the motivating issues common in vocational EFL settings. This research emphasizes the critical role of fostering a collaborative online learning community to enhance students' performance, consequently improving their engagement, consistent with the findings of [108–110]. This addressed as the principal impediment for vocational students: lack of motivation and engagement rather than cognitive incapacity. Interviews indicated that students had greater experiences of posing questions online compared to face-to-face classroom settings. That was because the CoI-based blended learning environment enabled students to share feelings, challenges, and achievements, thereby cultivating a supportive environment [26]. Such feeling of community is vital for enabling students to feel welcomed, supported, and accepted, hence increasing their VTS. These conclusions are consistent with the findings of [111–113]. CP was developed through well-organized exercises that fostered advanced engagement with vocabularies. According to [114], CP improves learner interest and active participation in online EFL discussions. Community members were urged to participate in peer interaction, problem-solving, inquiry, proposing solutions, and collaboration, thereby cultivating a flexible learning environment. Problem-based scenarios necessitated the contextual application of newly acquired vocabulary, fostering the desirable difficulty that has been demonstrated to improve long-term memory [115]. Metacognitive reflection exercises, in which students recorded and examined their vocabulary acquisition techniques, leveraged recent findings indicating that explicit strategy instruction can significantly aid vocational learners [116]. Furthermore, the components of the flipped classroom facilitated enhanced cognitive engagement during in-person sessions, aligning with recent research indicating that prior digital exposure liberates cognitive resources for advanced application in face-to-face classrooms [117].

The second sub-research question indicated a notable increase in the experimental group's VLM score of a 0.38-point improvement together with a medium effect size. This transition, as seen by [118], indicates a progression of students' VLM from a Moderate level to a Motivated level. The enhancement of students' VLM scores can be systematically elucidated by the CoI framework's three presences which fostered students' fundamental psychological needs of autonomy, competence, and relatedness [47]. First, by integrating well-crafted decision-making frameworks into blended learning, TP promoted students' autonomy. In this study, TP offered autonomous support via three processes, in accordance with [119]: 1) Self-directed vocabulary modules enabling students to follow their own learning pace, 2) optional challenge tasks for advanced learners, and 3) customized learning path generated by the Chaoxing Learning APP algorithm. Competence was developed through the immediate feedback from the teacher and system. Recent neurocognitive studies indicate when learners proficiently acquire new vocabularies, their brain's reward circuitry can be stimulated [120]. The varied assessment techniques in accordance with [121], such as peer evaluations and digital badges, established visible competence progression. It is essential for sustaining motivation in vocational students who frequently question their language learning capabilities. This study confirmed the findings of [122] that SP significantly affects learners' EFL motivation, engagement, and overall learning outcomes in blended or online language learning environments. SP markedly improved relatedness by organized peer interactions. Collaborative contexts and peer interactions enhanced competence development through modeling, feedback, and shared challenges. This consequently enhanced VLM, as learners who perceived themselves as competent and socially supported demonstrated increased persistence [47]. In addition, digitally-mediated cooperation can overcome the social anxiety commonly found in vocational classrooms. CP enhanced all three psychological demands via cognitively stimulating language exercises. In alignment with [27,123], this CoI-based integrated learning environment offers a forum for students to exchange their perspectives on vocabulary acquisition and engage in critical discourse. Students can express difficulties, seek knowledge, comprehend essential issues through the integration of existing information, and subsequently formulate solutions to tackle those challenges. Autonomy was augmented by problem-based learning scenarios that permitted several solution pathways. The CP features unexpectedly enhanced relatedness, as students identified common difficulties and solutions during collaborative meaning-making activities [124]. Tasks were calibrated to balance challenge and skill, avoiding both oversimplification and excessive difficulty. These

carefully scaffolded exercises fostered students' competence. Metacognitive reflection exercises further targeted vocational students' self-efficacy, with recent evidence showing such activities rebuild confidence in struggling learners. This sense of ownership and participation motivates students to engage more actively in their studies, in accordance with [80].

The results from the second research question indicated that students have a favorable and preferred perception of this CoI-based blended learning approach. The advantages of this CoI-based blended learning approach were revealed nine themes and 27 sub-themes from TP, SP and CP. In line with [110], when educators are adept at using technology resources, students' learning experiences are markedly enhanced. Align with [125,126], professional instruction, prompt feedback, varied evaluations, and consideration of students' emotions cultivated students' positive perceptions of TP. Furthermore, in accordance with [81] trust was fostered through collaboration, communication, and dialogue, which enhanced the development of confidence in self-expression. Students and teachers can openly discuss and express their authentic feelings through various channels, including online forums on the Chaoxing Learning App and face-to-face interactions. All these enable students' positive perceptions of SP. Moreover, the metacognitive components of this CoI-based blended learning environment effectively cultivated students' curiosity, strategic awareness and problem-solving skills [127], the enhanced CP helped address the recognized deficiency in the learning skills of vocational students.

Notwithstanding the favorable opinions expressed by the students, the implementation of this CoI-based blended learning approach in practical teaching also poses challenges. For example, students struggled to maintain sustained concentration during class. They also faced difficulties resisting distractions, particularly from mobile phone usage. Additionally, many found it challenging to avoid off-task conversations with peers. These issues were linked to underdeveloped self-regulation and autonomy skills. Therefore, students suggested that the teacher should give more supervision during the learning process in the future. In the same way, past studies [128–130] all noted that structured and supportive learning environments contribute to better academic resilience. Teachers ought to provide guiding roles by offering helpful feedback, impartial observations, and constructive views during tasks. Furthermore, students noted that the Chaoxing Learning App is inadequate for facilitating CoI-based blended learning. Future learning processes require a variety of supplementary digital tools that are engaging, valuable, user-friendly, and effective for interaction with instructors, peers, and content. In this way, instructors are able to develop more dynamic, efficient, and equitable language learning experiences [89]. Besides, some students believed that certain functions of the Chaoxing Learning App might be enhanced to increase its convenience. In the future, it should facilitate students' access to and download of courseware and instructional materials from any location at any time, while also allowing them to structure their own learning process. To address this problem, [72] suggested that the use of AI-driven platforms can facilitate more tailored and adaptive assessment experiences that correspond with the requirements of a contemporary, blended educational framework. What's more, the CoI-based blended learning approach also faces challenges in low-tech vocational settings. Dindar et al.'s [131] suggestion of using fundamental mobile platforms to act as alternatives to more advanced systems can be considered. For example, in high-tech regions, AI-powered tutoring system and VR simulations can be applied; however, in low-tech areas, We-chat group and Tencent QQ group can be used instead.

The thorough examination of students' VTS and VLM enhancement provided significant insights into the effectiveness of the CoI-based blended learning paradigm. The conceptual framework (Fig 1) synthesizes quantitative data from Research Question 1 and qualitative findings from Research Question 2, providing practical guidance for EFL educators aiming to apply these instructional strategies within Chinese secondary vocational education system. The related guidelines see appendices 1 and 2.

## Conclusions, limitations and recommendations

The findings of this study opened new opportunities for the investigation of effective ways of cultivating students' language skills in blended learning environments from the lens of CoI framework. By following these recommendations, students may be able to acquire English more effectively, particularly regarding vocabulary and vocabulary learning motivation. It is

hoped that the findings of this study might shed some light on influencing the mindsets of educators and content producers in order to enhance students' educational experiences and existing educational systems and procedures in terms of breadth, depth, efficacy, and efficiency. Moreover, this study provides EFL teachers with a CoI-based Blended Learning Conceptual Framework that may be used to design English language courses by incorporating digital tools that enhance students' Vocabulary Test Scores and Vocabulary Learning Motivation.

It is noteworthy that the present study is not without limitations. First, according to the characteristics of vocational majors, a limitation in terms of the gender imbalance among students exists in this study. Future studies could purposefully select gender-balanced cohorts from a variety of vocational disciplines to examine the effectiveness of the CoI framework in order to address these discrepancies. Although the sample's gender imbalance (81.8% male) reflects the demographic reality of Chinese vocational schools, it restricts the possibility to assess the potential influences of gender differences on CoI-based blended learning. Therefore, research on the effectiveness of this approach should focus on socioeconomic or gender-based demographics. Second, due to time constraints, this study was designed as a

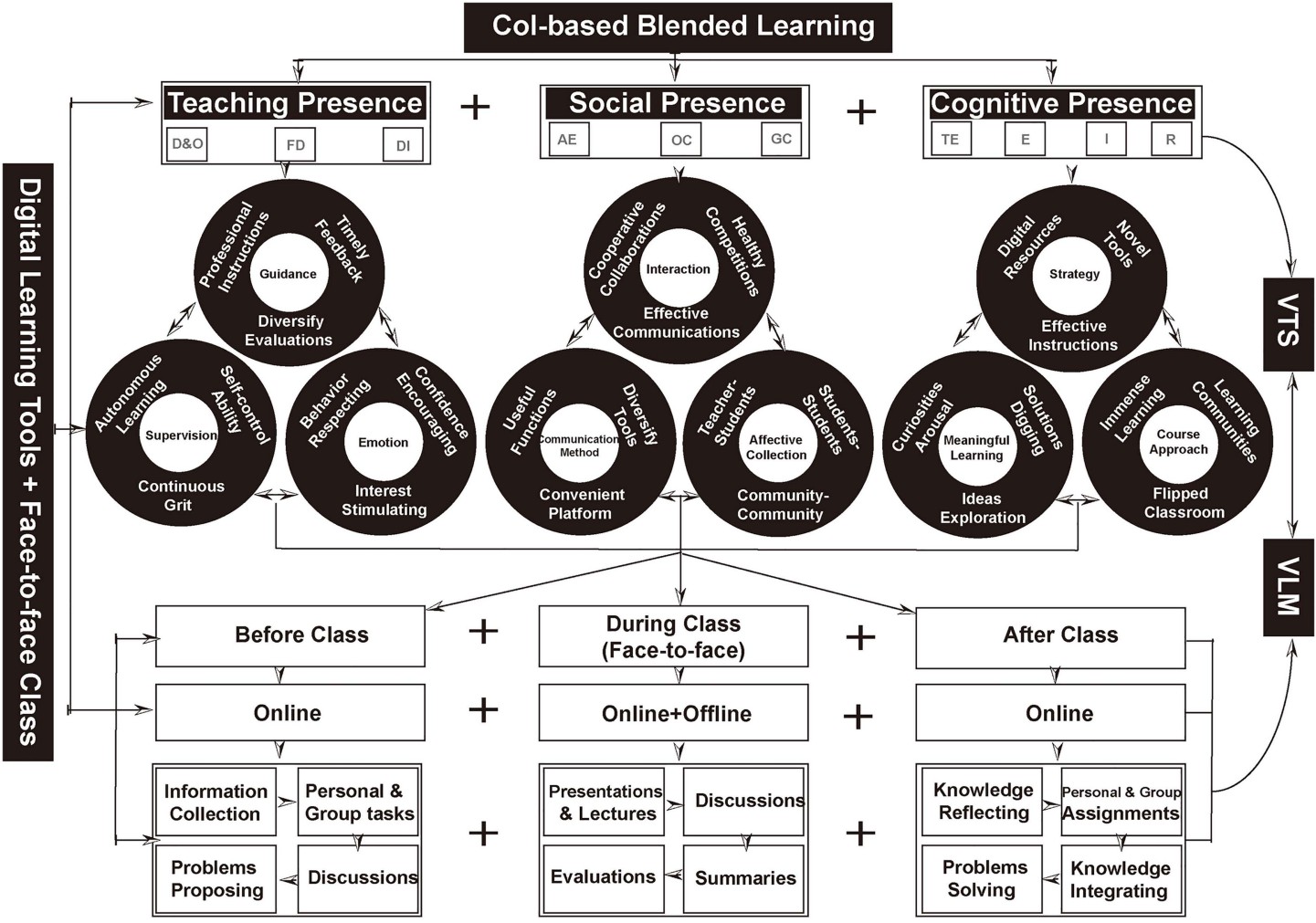

**Fig 1. Conceptual framework for enhancing students VTS and VLM.** *Teaching Presence: D&O: Design & Organization; FD: Facilitating Discourse; DI: Direct Instruction; Social Presence: AE: Affective Expression; OC: Open Communication; GC: Group Cohesion; Cognitive Presence: TE: Triggering Event; E: Exploration; I: Integrate: R: Resolution.

quasi-experiment with a duration of 13 weeks, to enhance analytical rigor, future research should incorporate longitudinal methodologies to explore causal relationships more robustly and validate findings on vocabulary retention and motivation. Third, other potentially significant factors, such as learner attitudes, autonomous learning behaviors, academic achievement, grit, and emotional regulation, were not explored in this study because it concentrated largely on vocabulary acquisition and vocabulary learning motivation within the CoI framework. Future studies are recommended to explore these variables from the perspective of CoI framework.

## Supporting information

**S1 File. Appendices-Guidelines for the framework.**
(DOCX)

**S2 File. Data of this study.**
(ZIP)

## Author contributions

**Data curation:** Qiu Chuane.

**Formal analysis:** Qiu Chuane.

**Methodology:** Qiu Chuane.

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
