## [Decision Letter · Decision Letter 0]

15 Apr 2025

PONE-D-25-04072Conceptual framework of applying CoI-based blended learning approach to enhance students’ vocabulary and vocabulary learning motivationPLOS ONE

Dear Dr. Chuane,

Thank you for submitting your manuscript to PLOS ONE. After careful consideration, we feel that it has merit but does not fully meet PLOS ONE’s publication criteria as it currently stands. Therefore, we invite you to submit a revised version of the manuscript that addresses the points raised during the review process.

We look forward to receiving your revised manuscript.

Kind regards,

Muhammad Zammad Aslam

Academic Editor

PLOS ONE

2. In the online submission form, you indicated that [All relevant data will be provided by the author on requrest.].

Additional Editor Comments:

1. Literature Review

Areas for Improvement:

- Digital Transformation and Assessment: Consider incorporating recent perspectives on digital transformation in EFL to strengthen the rationale for your use of blended learning platforms. For instance:

- Al Fraidan & Alaliwi (2024). Digital Transformation for Sustainable English Language Learning

- Al Fraidan (2025). Evaluating Lexical Competency in Saudi Arabia’s Hybridized EFL Ecosystem

- Learner Engagement and Motivation: The manuscript would benefit from integrating research on learner-centered practices in digital EFL contexts. In particular, the following work aligns closely with your study's emphasis on learner motivation and feedback responsiveness within digital pedagogies:

> Rasool, U., Qian, J. & Aslam, M. Z. (2024). Understanding the significance of EFL students’ perceptions and preferences of Written Corrective Feedback. Sage Open, 14(2), 1–11. https://doi.org/10.1177/21582440241256562

- Theoretical Anchoring: Incorporate Self-Determination Theory (Deci & Ryan) as a motivational lens to further explain how CoI elements (especially social and cognitive presence) relate to autonomy, competence, and relatedness in learning.

- Wording Clarity: Sentences like “students were frequently provided with voluminous paper-based vocabulary books…” could be revised to:

> “Traditionally, students have relied on extensive paper-based vocabulary exercises, a method that often reduces engagement and limits contextualized learning opportunities.”

2. Methodology and Analysis

Areas for Improvement:

- Group Assignment and Sampling: Clarify how experimental and control groups were matched or controlled for equivalence. A sample addition:

> “Using purposive sampling, both groups were matched based on baseline vocabulary scores and demographics, with pre-test homogeneity confirmed (t(84) = 0.20, p = 0.83).”

- Mapping Intervention to CoI: While Table 1 lists CoI elements and activities, the manuscript would benefit from a visual flowchart or timeline mapping phases of the intervention explicitly to CoI constructs (e.g., group tasks = social presence; teacher feedback = teaching presence).

- Instrumentation: Further details on validation of the vocabulary test and VLM questionnaire (e.g., pilot testing, expert review) would improve rigor.

- Qualitative Coding: Thematic/categorisation analysis using open, axial, and selective coding is well-described. Consider providing one example of a coding decision, for instance, you should make a section 'Trustwothiness', (please see for theoratical understanding of Trustworthiness Graneheim, U. H., & Lundman, B. (2004). Qualitative content analysis in nursing research: concepts, procedures and measures to achieve trustworthiness. Nurse Education Today, 24(2), 105-112. https://doi.org/10.1016/j.nedt.2003.10.001 and to add recent reference please see section 4.2 Trustworthiness of Alyaqoub, R., Alsharairi, A. & Aslam, M. Z. (2024). Elaboration of Underpinning Methods and Data Analysis Process of Directed Qualitative Content Analysis for Communication Studies. Journal of Intercultural Communication, 24(2), 108-116. https://doi.org/10.36923/jicc.v24i2.573 ) and mention whether intercoder reliability was checked or why not.

- Statistical Assumptions: Add notes on testing for normality (e.g., Shapiro–Wilk) and clarify whether assumptions for parametric tests were met.

- Effect Size Interpretation: Though eta squared is reported, please explain its educational significance. For example, what does η² = 0.232 mean in practice for vocabulary learning outcomes?

3. Results and Discussion

Suggestions:

- Educational Impact of Gains: Elaborate on what a ~9-point gain in VTS means pedagogically (e.g., is it equivalent to one academic level?).

- Qualitative Summary Table: Include a table linking themes to representative student quotes to enhance clarity and synthesis.

- Comparative Framing: Relate your findings more explicitly to previous studies such as Chen et al. (2021) and Garrison (2017). Do your results reinforce or extend their conclusions?

- Scalability: Comment on the practicality and potential scalability of the CoI-based model in other vocational or low-tech environments.

---

4. Figures and Framework

Suggestions:

- Annotate the framework with possible real-world actions. For example, under “Teaching Presence,” add examples like “use of recorded feedback” or “student-led microteaching.”

- Offer low-tech adaptations for under-resourced settings.

5. Language and Formatting

Needs Attention:

- Grammar and phrasing require minor editing. For instance:

- “The lacking amount of vocabulary…” → “The lack of vocabulary…”

- “Their VTS and VLM…” → Use full terms on first use in each section.

- Ensure verb tenses are consistent, particularly in the methodology and results sections.

- APA/PLOS formatting for p-values and statistical terms should be checked.

6. Limitations and Future Research

Suggestions:

- Propose longitudinal studies on vocabulary retention and motivation.

- Suggest cross-discipline comparisons or cross-platform CoI applications.

- Discuss gender-based or socio-economic implications in vocational settings.

Reviewers' comments:

Reviewer's Responses to Questions

**Comments to the Author**

1. Is the manuscript technically sound, and do the data support the conclusions?

Reviewer #1: Yes

Reviewer #2: Partly

Reviewer #3: Yes

2. Has the statistical analysis been performed appropriately and rigorously? 

Reviewer #1: Yes

Reviewer #2: No

Reviewer #3: Yes

3. Have the authors made all data underlying the findings in their manuscript fully available?

Reviewer #1: Yes

Reviewer #2: No

Reviewer #3: Yes

4. Is the manuscript presented in an intelligible fashion and written in standard English?

Reviewer #1: Yes

Reviewer #2: Yes

Reviewer #3: Yes

5. Review Comments to the Author

Reviewer #1: The manuscript is well organized and well written. The discussion section is strong with previous studies comparisons and contrasts however i would like to suggest if figure 5.1 can be re designed or replace as it seems little unclear at present.

Reviewer #2: The manuscript investigates how a Community of Inquiry (CoI)–based blended learning approach affects vocabulary test scores (VTS) and vocabulary learning motivation (VLM) among Chinese secondary vocational EFL students. Using an explanatory sequential mixed methods design, the study compares an experimental group (using the CoI-based blended learning intervention via the Chaoxing Learning App) with a control group (traditional face-to-face instruction). Quantitative data (pre/post vocabulary tests and VLM questionnaires) are analyzed with paired and independent t-tests, and qualitative data from semi-structured interviews are thematically coded. The research questions focus on both the quantitative impact and the students’ perceptions of the intervention.

⸻

Major Comments

1. Literature Review – Theoretical Integration and Relevance

• Current Strengths:

The literature review is extensive and well structured, covering the key components of the CoI framework (teaching, social, and cognitive presence) and linking these to vocabulary learning challenges in the EFL context. The discussion on how blended learning can overcome the limitations of rote vocabulary teaching is useful.

• Areas for Improvement:

• Integration of Digital Transformation Perspectives:

Although the review discusses blended learning and CoI, it would benefit from more discussion of the digital transformation aspect. For example, your work:

• Al Fraidan, A. & Alaliwi, M. (2024). Digital Transformation for Sustainable English Language Learning: Insights from Saudi Arabia and Global Perspectives

could be cited to provide a cross-cultural or global perspective on how digital tools enhance language learning, thereby reinforcing the rationale for using a technology-based platform like Chaoxing Learning App.

• Vocabulary Assessment and Motivation:

The review would also be strengthened by linking vocabulary testing and motivation with recent findings. You might include:

• Al Fraidan, A. (2025). Evaluating Lexical Competency in Saudi Arabia’s Hybridized EFL Ecosystem: A Taxonomic Exploration of Vocabulary Assessment Modalities and the Imperative for AI-Enhanced Adaptive Testing

to support the discussion of vocabulary assessment challenges and the need for innovative instructional designs.

• Excerpts for Revision:

For instance, the sentence:

“Students were frequently provided with voluminous paper-based vocabulary books and subjected to the laborious process of passive learning.”

could be revised for clarity and depth to:

“Traditionally, students have relied on extensive paper-based vocabulary exercises, a method that often reduces engagement and limits the opportunity for contextualized learning. Recent studies (e.g., Al Fraidan, 2025) underscore the need for adaptive and interactive assessment modalities to enhance both vocabulary acquisition and learner motivation.”

2. Methodology – Design, Intervention, and Data Analysis

• Design and Sampling:

• Clarification Needed:

While the use of a quasi-experimental design is appropriate, further detail is needed on the sampling and group assignment process. Explain how you ensured the experimental and control groups were comparable in terms of English proficiency and demographic factors. Specify whether random assignment (or matching) was used.

• Suggestion:

Include a brief statement on how you minimized potential confounders. For example:

“Using purposive sampling, both groups were matched on baseline vocabulary scores and demographic characteristics, and pre-test homogeneity was confirmed via an independent samples t-test (t(84)=0.20, p=0.83).”

• Intervention Description:

• Enhancement of Details:

The intervention plan is described, but the link between CoI elements and specific blended learning activities can be clearer. For instance, the activities in Table 1 (elements, categories, and activities) should be explicitly tied back to the theoretical constructs of TP, SP, and CP.

• Suggestion:

Provide a flowchart or timeline that maps each intervention phase to the corresponding CoI element(s). Clarify how online discussions, quizzes, and group tasks contribute to cognitive and social presence.

• Data Collection and Instrumentation:

• Instrumentation Validity:

The researcher-developed vocabulary test and VLM questionnaire are mentioned, and Cronbach’s Alpha values are provided. However, additional details on the validation process (e.g., expert review, pilot testing) would strengthen the methodological rigor.

• Qualitative Data:

The description of the thematic analysis (using Open, Axial, and Selective Coding) is clear. Yet, consider providing examples of the coding process and how inter-coder reliability was established.

• Statistical Analysis:

• Appropriateness:

The use of paired and independent t-tests is appropriate for the pre/post comparisons. The reporting of effect sizes (Eta squared) is also commendable.

• Suggestions for Improvement:

• Report confidence intervals for the effect sizes to better interpret practical significance.

• Address whether the data met the assumptions of normality (e.g., via Shapiro–Wilk tests) and discuss any steps taken if assumptions were violated.

• Although the effect sizes for VTS improvements (η² = 0.232) and VLM (η² = 0.092 and 0.279) are reported as “moderately weak,” a discussion of what these values imply in an educational context would be beneficial.

3. Results and Discussion

• Results Presentation:

• The quantitative results are clearly presented in tables with appropriate statistical values. However, the narrative could be improved by integrating the numbers with practical implications.

• For example, in Table 4.5, you report that experimental group VTS increased from 15.64 to 25.43 (t(43) = -8.627, p = 0.000). The discussion should explain what a 9-point increase on a 40-point scale means for student learning outcomes.

• Qualitative Findings:

• The qualitative findings are detailed, with themes related to Teaching Presence, Social Presence, and Cognitive Presence well described. Consider adding a summary table that links themes with representative quotes to improve clarity.

• Discussion of Findings:

• The discussion appropriately links the findings back to the CoI framework. However, it would benefit from a more critical comparison with previous studies. For example, discuss how your findings align with or differ from those reported by, say, Chen et al. (2021) or Garrison (2017), and what this means for EFL pedagogy.

• Additionally, explicitly discuss the practical significance of the blended learning intervention in terms of potential scalability in vocational settings.

⸻

Minor Comments and Technical Corrections

• Language and Style:

• Several sentences are lengthy and could be streamlined for clarity. For example, revise:

“The teacher provided detailed feedback and systematic information about their presentation, allowing students to actively engage in discussions and develop their understanding through peer and teacher interaction.”

to a more concise version.

• Check for minor typographical errors (e.g., “is there any improvement of their VTS and VLM” should be “if there is any improvement”).

• Formatting:

• Tables and figures should be uniformly formatted. Ensure that table captions are clear and that all abbreviations (e.g., VTS, VLM) are defined in each table.

• In the literature review, ensure that all in-text citations are consistent and correctly formatted.

• Excerpts to be Corrected:

• Example Excerpt (Literature Review):

“Over the past few years, blended learning has become increasingly popular …”

Suggest revising to:

“In recent years, the adoption of blended learning has surged, owing to its flexibility and the integration of digital tools that foster both independent and collaborative learning (Cronje, 2020; Polakova & Klimova, 2022).”

• Example Excerpt (Methodology):

“The Vocabulary Test compromises 40 multiple-choice questions…”

Correct to:

“The Vocabulary Test consists of 40 multiple-choice questions…”

⸻

To further enrich the literature review and strengthen the manuscript’s theoretical grounding, I recommend citing some of these relevant works:

• Digital Transformation in Language Learning:

• Al Fraidan, A. & Alaliwi, M. (2024). Digital Transformation for Sustainable English Language Learning: Insights from Saudi Arabia and Global Perspectives.

This work can be used to highlight the broader context of digital transformation and blended learning innovations.

• Vocabulary Assessment and Adaptive Testing:

• Al Fraidan, A. (2025). Evaluating Lexical Competency in Saudi Arabia’s Hybridized EFL Ecosystem: A Taxonomic Exploration of Vocabulary Assessment Modalities and the Imperative for AI-Enhanced Adaptive Testing.

This article would support the discussion on innovative vocabulary assessment methods and the need for adaptive strategies in EFL contexts.

• Motivation and Test Anxiety in Language Learning:

• Al Fraidan, A. & AlDossari, M. S. A. (2025). Navigating the Lexical Labyrinth: Vocabulary Test Anxiety, Teacher Strictness, and Strategic Mastery in ESL Assessment.

This study provides insights into the affective components of language learning which could complement your discussion of vocabulary learning motivation.

⸻

Recommendation

Decision: Major Revision with Conditional Acceptance

While the manuscript demonstrates promise and addresses an important topic in EFL education through a CoI-based blended learning framework, the following revisions are necessary before acceptance:

1. Expand and integrate the literature review with additional references (including your own relevant work) to more robustly justify the theoretical framework and the need for blended learning interventions in vocabulary acquisition.

2. Provide additional methodological details regarding group assignment, instrument validation, and control of extraneous variables.

3. Enhance the statistical reporting by including tests for normality, confidence intervals for effect sizes, and a discussion of practical significance.

4. Clarify the qualitative data analysis with more detailed descriptions of the coding process and inter-coder reliability.

5. Revise language for clarity, shorten overly long sentences, and standardize formatting across tables and figures.

I recommend that the authors address these points in a revised submission. With these improvements, the manuscript will make a significant contribution to our understanding of CoI-based blended learning in EFL settings and is likely to be acceptable for publication.

⸻

I trust these comments and suggestions will assist in further refining the manuscript.

Reviewer #3: Dear Author,

Your manuscript presents a timely and valuable contribution to the field of EFL education, particularly in the context of secondary vocational schools in China. I commend your use of the Community of Inquiry (CoI) framework and the blended learning model, both of which are pedagogically relevant and underexplored in this setting.

Strengths:

Well-grounded in theory, particularly CoI and blended learning pedagogies.

Effective use of mixed methods that adds depth to your findings.

Clear evidence of statistical and practical significance in improving students’ vocabulary scores and motivation.

Qualitative insights enrich the narrative and give voice to student experiences.

Suggestions for Improvement:

Language and Style:

Please consider a professional language edit. A few examples for revision:

“The lacking amount of vocabulary…” → “The lack of vocabulary…”

“their VTS and VLM” → better to spell out at first mention in each section.

Verb tense consistency in the discussion and methodology needs attention.

Data Availability:

Though you mentioned data is available upon request, PLOS encourages open-access sharing of datasets. Please consider uploading your anonymized data to a repository like OSF and linking it in your manuscript.

Theoretical Implications:

The discussion would benefit from deeper theoretical integration—for example, explicitly linking student motivation to the CoI's presences, or citing motivational frameworks like Self-Determination Theory to anchor the findings.

Limitations and Future Research:

You identified gender imbalance and sample size/location as limitations. You might further suggest longitudinal studies to explore retention of vocabulary and motivation over time, or comparative studies across disciplines.

Figures & Framework (Fig 5.1):

Your conceptual framework is a useful synthesis. To enhance its practical value, consider explaining how teachers might operationalize each element in low-tech or resource-constrained classrooms.

6. PLOS authors have the option to publish the peer review history of their article (what does this mean? ). If published, this will include your full peer review and any attached files.

**Do you want your identity to be public for this peer review?** For information about this choice, including consent withdrawal, please see our Privacy Policy .

Reviewer #1: **Yes: ** Ushba Rasool

Reviewer #2: No

Reviewer #3: **Yes: ** Dr. Salahud Din Abdul-Rab

---

## [Author Response · Author response to Decision Letter 1]

16 Jun 2025

Dear Editor and Reviewers,

The revised details please see the file of "Response to editor and reviewers".

---

## [Decision Letter · Decision Letter 1]

28 Jul 2025

Conceptual framework of applying CoI-based blended learning approach to enhance students’ vocabulary and vocabulary learning motivation

PONE-D-25-04072R1

Dear Dr. Chuane,

We’re pleased to inform you that your manuscript has been judged scientifically suitable for publication and will be formally accepted for publication once it meets all outstanding technical requirements.

Kind regards,

Muhammad Zammad Aslam, Ph.D.

Academic Editor

PLOS ONE

Additional Editor Comments (optional):

Reviewers' comments:

Reviewer's Responses to Questions

**Comments to the Author**

1. If the authors have adequately addressed your comments raised in a previous round of review and you feel that this manuscript is now acceptable for publication, you may indicate that here to bypass the “Comments to the Author” section, enter your conflict of interest statement in the “Confidential to Editor” section, and submit your "Accept" recommendation.

Reviewer #2: All comments have been addressed

2. Is the manuscript technically sound, and do the data support the conclusions?

Reviewer #2: Yes

3. Has the statistical analysis been performed appropriately and rigorously? 

Reviewer #2: Yes

4. Have the authors made all data underlying the findings in their manuscript fully available?

Reviewer #2: Yes

5. Is the manuscript presented in an intelligible fashion and written in standard English?

Reviewer #2: Yes

6. Review Comments to the Author

Reviewer #2: (No Response)

7. PLOS authors have the option to publish the peer review history of their article (what does this mean? ). If published, this will include your full peer review and any attached files.

**Do you want your identity to be public for this peer review?** For information about this choice, including consent withdrawal, please see our Privacy Policy .

Reviewer #2: No
